


# Reanalysis of NOAA $H_2$ observations: implications for the $H_2$ budget

Fabien Paulot[1], Gabrielle Pétron[2,3], Andrew M. Crotwell[2,3], and Matteo B. Bertagni[4,5]

[1]Geophysical Fluid Dynamics Laboratory, National Oceanic and Atmospheric Administration, Princeton, NJ, USA
[2]Cooperative Institute for Research in Environmental Sciences, University of Colorado Boulder, Boulder, CO, USA
[3]Global Monitoring Laboratory, National Oceanic and Atmospheric Administration, Boulder, CO, USA
[4]High Meadow Environmental Institute, Princeton University, Princeton, NJ, USA
[5]Department of Civil and Environmental Engineering, Princeton University, Princeton, NJ, USA

**Correspondence:** Fabien Paulot (fabien.paulot@noaa.gov)

**Abstract.** Hydrogen ($H_2$) is being considered for many applications as an alternative to fossil fuels. Robust assessment of the climate implications of increased $H_2$ usage in the global economy is partly hindered by uncertainties in its biogeochemical cycle. Here we use NOAA $H_2$ dry air mole fraction observations from air samples collected from ground-based and ship platforms from 2010 to 2019 to evaluate the representation of $H_2$ in the NOAA GFDL-AM4.1 atmospheric chemistry-climate model. We find that the model captures the observed interhemispheric gradient well but underestimates the surface concentration of $H_2$ by about 10 ppbv. Observations show a 1-2 ppbv/year mean increase in surface $H_2$ at background stations, while the simulated $H_2$ exhibits no significant change over the 2010–2019 period. We show that this model bias is primarily driven by the estimated decrease of anthropogenic emissions, mostly from transportation, and that including leakage from $H_2$-producing facilities can improve the simulated trend. We find that changes in soil moisture, soil temperature, and snow cover likely increase the magnitude and modify spatial distribution of the soil sink, the most important removal mechanism for atmospheric $H_2$. However, the magnitude and even the sign of such changes is uncertain due to fundamental gaps in our understanding of $H_2$ soil removal, such as the minimum soil moisture for $H_2$ soil uptake. We show that the observed meridional gradient of $H_2$ mixing ratio and its seasonality provide important constraints to test and refine parameterizations of $H_2$ soil removal.

## 1 Introduction

Increased hydrogen ($H_2$) usage has been proposed as a strategy to reduce the carbon intensity of many sectors of the economy that are difficult to electrify (Hydrogen Council, 2017; da Silva Veras et al., 2017; Staffell et al., 2019; Abe et al., 2019; Dawood et al., 2020). The climate benefits of greater hydrogen usage depend primarily on the $H_2$ production pathway. Current hydrogen production is dominated by steam reforming of methane in natural gas (Holladay et al., 2009; International Energy Agency, 2019), a process that is very carbon intensive (Howarth and Jacobson, 2021). Carbon capture can reduce $CO_2$ emissions associated with hydrogen production but the increased demand for $CH_4$ may offset much of the expected climate benefits of increased $H_2$ usage (Howarth and Jacobson, 2021; Ocko and Hamburg, 2022; Bertagni et al., 2022; Hauglustaine et al., 2022). Alternative production pathways such as renewable-based electrolytic $H_2$ have been estimated to provide large and




rapid reductions in radiative forcing (Hauglustaine et al., 2022) and considerable investments have been devoted to reducing their cost (International Energy Agency, 2022). Furthermore, evidence of high concentrations of $H_2$ in many different geologic

environments (Zgonnik, 2020) have spurred interest in the potential of naturally-occurring $H_2$ as a new primary energy source (Prinzhofer et al., 2018; Lapi et al., 2022).

Assessing the potential climate benefits of greater $H_2$ usage also requires us to quantify the environmental impact of the atmospheric release of $H_2$. Recent studies indicate that $H_2$ has a global warming potential (100 years) of $\simeq 10$ (Derwent, 2022; Warwick et al., 2022; Hauglustaine et al., 2022). The radiative impact of $H_2$ is indirect, reflecting the increase in $CH_4$, $O_3$, and

stratospheric water vapor associated with its photooxidation (Derwent et al., 2001; Paulot et al., 2021). $H_2$ photooxidation is estimated to account for 20-30% of the overall sink of $H_2$, which is dominated by soil uptake (Ehhalt and Rohrer, 2009). As a result, the soil sink tends to reduce the indirect radiative forcing of $H_2$.

We recently presented an assessment of $H_2$ indirect radiative forcing using the Geophysical Dynamics Laboratory (GFDL) AM4.1 model (Paulot et al., 2021). Here, we leverage the recently completed recalibration of $H_2$ measurements collected

by NOAA Global Monitoring Laboratory. This monitoring network provides additional spatial coverage that complements other existing networks (AGAGE (Prinn et al., 2018), CSIRO (Francey et al., 2003)). Here, we first describe and evaluate the representation of $H_2$ in the GFDL-AM4.1 global chemistry-climate model, focusing on changes in $H_2$ over the 2010–2019 period. We then evaluate the impact of $H_2$ anthropogenic sources and soil removal on the simulated seasonality and trends of $H_2$.

## 2 Methods

### 2.1 Observations

NOAA Global Monitoring Laboratory (GML) provides long-term monitoring of long-lived greenhouse gases and other trace species. The NOAA GML Global Cooperative Air Sampling Network is a partnership between GML and many outside organizations and individual volunteers to collect discrete air samples approximately weekly from 60+ globally distributed sites.

These sites are often situated to collect air representative of large regional air masses. Priorities are placed on sites where opportunities exist for local support which can be maintained over long (decadal) time scales. The discrete air samples are collected weekly in pairs of 2 L glass flasks and are returned to GML for measurements of multiple species on central measurement systems thus providing a high level of consistency across the globally distributed network. (add references)

GML measurements of $H_2$ in the discrete air samples began in the late 1980's as an opportunistic measurement associated

with the analytical technique then used for measuring atmospheric carbon monoxide (CO). To facilitate these $H_2$ measurements, NOAA/GML developed an in-house $H_2$-in-air reference scale based on a few gravimetric standards (the latest iteration named H2-X1996). This reference scale was not stable over time and introduced significant time-dependent measurement errors. GML recently converted part of the historical $H_2$ measurement records to the $H_2$ calibration scale recommended by the World Meteorological Organization (WMO/MPI H2-X2009) maintained by Max Planck Institute (MPI) in Jena, Germany (Jordan

and Steinberg, 2011). Measurements since approximately 2010 have been reprocessed onto the MPI scale to remove the biases





inherent in the NOAA X1996 scale. (Pétron et al, in preparation). NOAA reprocessed $H_2$ data since 2010 is consistent with other measurement labs which maintain tight connections to the MPI central calibration facility. However, earlier NOAA data that remains on the obsolete NOAA X1996 scale is known to be biased relative to the later NOAA data and to other monitoring programs.

Here, we only consider ground stations from the NOAA cooperative air sampling network with at least 96 distinct monthly observations over the 2010-2019 period (80% coverage). Ship-based observations are binned in $4°x4°$ regions and we only consider regions with at least 40 observations.

## 2.2   Global model

We use the GFDL Atmospheric Chemistry Model AM4.1 (Horowitz et al., 2020). AM4.1 includes a detailed representation
of $H_2$ (Paulot et al., 2021), which is briefly summarized here. This configuration will hereafter be referred to as BASE (Table 1). $H_2$ sources include both direct emissions and photochemical productions. Anthropogenic emissions of $H_2$ are estimated from CO emissions in the Community Emissions Data System (CEDS) v20210421 (O'Rourke et al., 2021) using time-invariant sector–specific emission ratios (Table A1). Biomass burning emissions are estimated using the Global Fire Emissions Database (GFED4s, van der Werf et al. (2017)) with emission factors from Akagi et al. (2011) and Andreae (2019). Marine (6 Tg/yr)
and terrestrial (3 Tg/yr) sources of $H_2$ are prescribed as a monthly climatology based on Paulot et al. (2021).

$H_2$ is also produced from the photolysis of formaldehyde ($CH_2O$). Formaldehyde sources are dominated by the oxidation of volatile organic compounds (VOCs) from anthropogenic (O'Rourke et al., 2021), biomass burning (van der Werf et al., 2017), and natural origins. Biogenic emissions of VOCs are prescribed as a monthly climatology (Granier et al., 2005), except for isoprene and terpenes, of which emissions are calculated using the Model of Emissions of Gases and Aerosols from Nature
(Guenther et al., 2012). Surface $CH_4$ is prescribed as a monthly latitudinal profile from observations up to 2014 (Meinshausen et al., 2017) and from the SSP1-2.6 scenario after 2015 (Meinshausen et al., 2020). We select this scenario as it tracks well the observed global $CH_4$ surface mixing ratio from the World Meteorological Organization Global Atmospheric Watch greenhouse gases observational network (WMO, 2021).

$H_2$ sinks include chemical oxidation by OH and $O(^1D)$, and soil uptake associated with microbial activity. In the BASE
configuration, the deposition velocity of $H_2$ ($v_d(H_2)$) over land is calculated following the parameterization of Ehhalt and Rohrer (2013) and depends on temperature, soil moisture (Ehhalt and Rohrer, 2013) and soil carbon (Khdhiri et al., 2015; Paulot et al., 2021). Here, we drive the BASE simulation with a monthly climatology of $v_d(H_2)$ calculated using monthly meteorological and soil outputs from the GFDL Earth System Model ESM4.1 over the 1989–2014 period (Dunne et al., 2020; Paulot et al., 2021).

In addition to the BASE configuration, we perform sensitivity simulations using a more comprehensive treatment of $H_2$ emissions (REVISED) and $H_2$ soil removal (REVISED_GLDAS, REVISED_GLDAS2). These configurations are described in sections 4.1 and 4.2 and summarized in Table 1.

The model is run from 2004 to 2019. Monthly sea surface temperature and sea ice concentration are from Rayner et al. (2003) and Taylor et al. (2000). Horizontal winds are nudged to 6-hourly horizontal winds from the National Center for Environmental



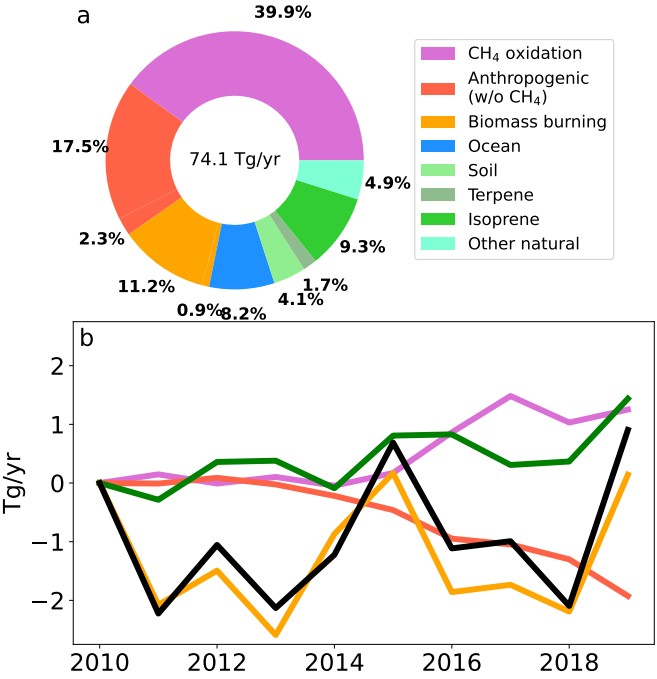

**Figure 1.** Global source of $H_2$ (black, panel a). Dotted wedges indicate photochemical sources. Panel b shows the changes in the magnitude of $H_2$ sources over the 2010–2019 period. For clarity, the green line denotes the combined change in $H_2$ emissions and photochemical production from natural sources (marine and soil emissions + BVOCs photooxidation).

Prediction (Kalnay et al., 1996). The model output is sampled at the time and location of the air sampling. To better quantify the drivers of the $H_2$ distribution and trend, we add five different tracers that represent $H_2$ associated with anthropogenic, marine, soil, and biomass burning direct $H_2$ emissions and $H_2$ produced by VOC oxidation.

## 3   Results

### 3.1   Global budget

Fig. 1a summarizes the simulated sources of $H_2$ associated with photochemical production (dots) and emissions (solid color). Over the 2010-2019 period, the average global simulated source of $H_2$ is 74.1±1 Tg/yr. The contribution of $CH_4$ oxidation is estimated by separately tracking the different $CH_4$ oxidation pathways that result in $H_2$ production. The contribution of other photochemical pathways is estimated by perturbing the associated precursor emissions by 10%.

$CH_4$ oxidation is the single largest source of $H_2$ (29.6 Tg/yr) accounting for $\simeq 40\%$ of the overall $H_2$ source and 2/3 of its photochemical source. This contribution is larger than estimated by Ehhalt and Rohrer (2009) (23 Tg/yr, 30% and 56% in





**Table 1.** Model configurations

| | **H₂ anthropogenic emission** | **H₂ natural emission** | **H₂ soil removal** |
|---|---|---|---|
| **BASE** | Time-invariant emissions factor (Paulot et al., 2021) | Ocean+Soil: Monthly climatology<br>Biomass burning: GFED | Monthly climatology $v_d(H_2)$<br>(Ehhalt and Rohrer, 2013; Paulot et al., 2021) |
| **REVISED** | Time-varying emissions factor<br>Appendix A1 | Ocean: Calculated from CO seawater distribution<br>Soil: Calculated from biological nitrification<br>Appendix A2<br>Biomass burning: same as BASE | Same as BASE |
| **REVISED_GLDAS** | same as REVISED | Same as REVISED | Daily $v_d(H_2)$ calculated from 3-hourly GLDAS soil moisture and temperature<br>Appendix B |
| **REVISED_GLDAS2** | same as REVISED | Same as REVISED | Same as REVISED_GLDAS with a lower HA-HOB water-activation threshold and canopy+litter resistance<br>Appendix B |



2005, respectively). Two factors contribute to this difference: a) greater oxidative flux of $CH_4$ (560 Tg/yr, $+\simeq 12\%$) and b) higher yield of $H_2$ from $CH_4$ oxidation (0.42 mol($H_2$)/mol($CH_4$) compared to 0.37 mol($H_2$)/mol($CH_4$)).

The second most important photochemical source of $H_2$ is the photooxidation of isoprene. Isoprene is primarily emitted from plant foliage and accounts for $\simeq 50\%$ of the global emissions of non-methane volatile organic carbon (NMVOC, Guenther et al. (2006)). We estimate that the oxidation of isoprene yields $\simeq 0.1$ mol($H_2$)/mol(C), which amounts to $\simeq 6.9$ Tg/yr or $\simeq 9\%$ of the overall source of $H_2$). The oxidation of other biogenic NMVOCs accounts for the majority of the remaining photochemical source of $H_2$ ($\simeq 4.9$ Tg/yr) with smaller contributions from the photooxidation of NMVOCS from anthropogenic (2.3%) and biomass burning (0.9%) origin. Anthropogenic activities are estimated to contribute over 40% of the overall $H_2$ source including 17.5% from direct emissions (associated with fossil fuel combustion), 2.3% from NMVOC oxidation, and 22% from $CH_4$. The $CH_4$ estimate is obtained by scaling the global source of $H_2$ from $CH_4$ by the estimated contribution of anthropogenic sources to $CH_4$ emissions (50-62% (Saunois et al., 2020)).

The simulated total source of $H_2$ changes little over the 2010–2019 period. The annual production of $H_2$ associated with the photooxidation of $CH_4$ and NMVOCs is 1.25 Tg/yr and 1.45 Tg/yr (0.95 Tg/yr from isoprene) greater in 2019 than in 2010, respectively. This increase is largely compensated by a decrease in emissions of $H_2$ associated with anthropogenic activities (-1.93 Tg/yr). As we will discuss in section 4.1, this decline is primarily driven by a decrease in anthropogenic CO emissions from the transportation sector and assuming the same behaviour for $H_2$ emissions. The interannual variability of the overall $H_2$ source over the 2010-2019 period is dominated by the variability of biomass burning emissions.

The overall lifetime of $H_2$ in the BASE configuration is 2.5 years. The lifetime of $H_2$ associated with anthropogenic emissions is 6% shorter due to their geographical distribution. Soil uptake is estimated to account for 71% of the overall $H_2$ sink.

## 3.2 Evaluation

Fig. 2 shows the average model bias against surface observations from NOAA GML. In the BASE configuration, AM4.1 underestimates $H_2$ at all stations, with greater biases over continental regions (Fig. 2). Correlations exceed 0.5 at more than 90% of background sites (square) but only 55% of continental sites. Fig. 2b shows that the magnitude of the pole to pole gradient ($\simeq 50$ ppbv) is well captured.

To examine differences between the model and observed seasonality, we first apply the Kmean++ clustering algorithm (Arthur and Vassilvitskii, 2007) to the observed $H_2$ monthly climatology. Since our focus is on the seasonality of $H_2$ we transform the monthly climatology of $H_2$ at each site such that it has a mean of 0 and a standard deviation of 1. Using the within-cluster sum of squares and the silhouette score, we find that the standardized $H_2$ observations can be well represented using 4 clusters. Fig. 3 shows the seasonality of the standardized $H_2$ concentration for each cluster (panel a) as well as their spatial distribution (panel b). Sites are found to cluster broadly by latitude based on the seasonality of $H_2$ with clusters 1, 2, 3, and 4 being comprised primarily of sites located in the Southern mid to high latitudes, Southern tropics, Northern subtropics, and Northern mid to high latitudes, respectively. The model captures the seasonality of $H_2$ well in the Southern Hemisphere (cluster 1) but peaks 1 to 3 months earlier than observations for clusters 2, 3 and 4. Fig. 3c shows the contribution of different sources of $H_2$ to the simulated seasonality of $H_2$ (inferred from the tagged $H_2$ tracers). The seasonal bias for cluster 2 is





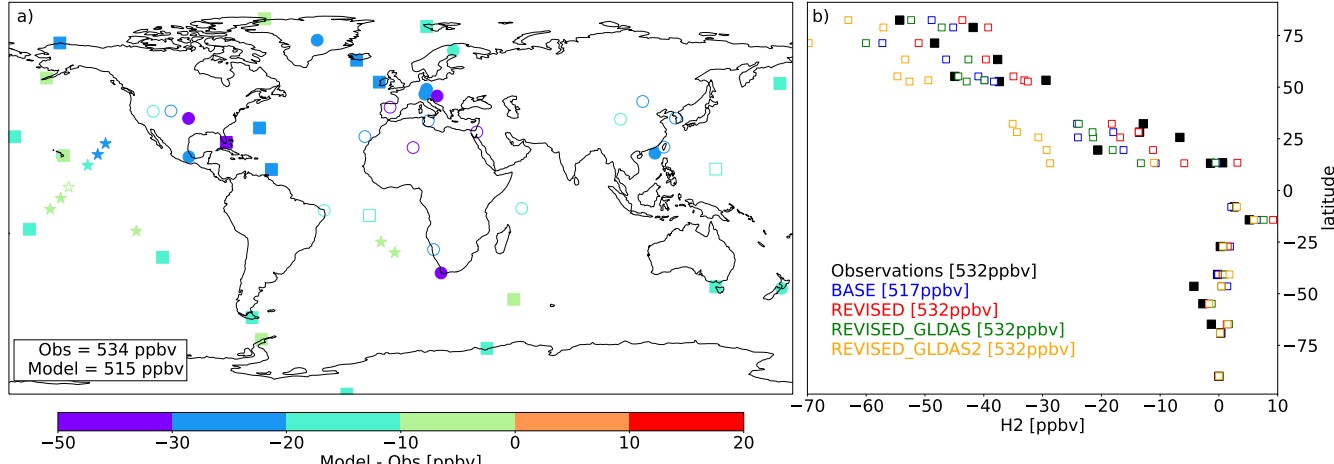

**Figure 2.** Mean model bias at individual sites for the BASE model configuration (a) over the 2009–2019 period. Filled symbols denote sites where the correlation between observed and simulated $H_2$ concentrations exceeds 0.5. Square and star symbols denote background sites and cruises, respectively. Panel (b) shows the observed and simulated difference in $H_2$ at background sites relative to $H_2$ mole fraction measured at the South Pole observatory. The average concentrations at background sites is indicated for each configuration in the legend.

primarily driven by $H_2$ emitted from biomass burning, which peaks $\sim$ 2 months earlier than observations. This delay may be associated with greater burning of woody material towards the end of the dry season, emitting more incompletely oxidized products such as $H_2$ (van der Werf et al., 2006). Fig. 3c also shows that the seasonal bias in clusters 3 and 4 may be associated with $H_2$ emitted by anthropogenic activities. As we will show in section 4.2, this seasonal bias may also reflect errors in the removal of $H_2$.

Fig. 4 shows $that H_2$ has increased at most sites with an average trend at background sites of 1.4±0.7 ppbv/yr over the 2010-2019 period with little variability with latitude. Trends are calculated using ordinary-least-square regression applied to the deseasonalized monthly $H_2$ concentrations. In contrast, no significant change in $H_2$ concentration is simulated in the BASE configuration (0.045±0.4 ppbv/yr at background sites).

In the Northern hemisphere, the lack of trend at background sites in the simulated $H_2$ concentration (Fig. 4c) reflects the cancellation between the increase of photochemically-produced $H_2$ and the decrease of $H_2$ emitted from anthropogenic sources. The simulated absolute trend in anthropogenic hydrogen is $\simeq$ 50% lower in the Southern Hemisphere relative to the Northern Hemisphere due to the higher relative areal density of anthropogenic sources in the Northern Hemisphere. In contrast, the change in photochemically-produced $H_2$ exhibits little variability with latitude and matches the observed trend well. The simulated trend also shows little latitudinal variation due to a decrease in $H_2$ from biomass burning in the Southern Hemisphere.



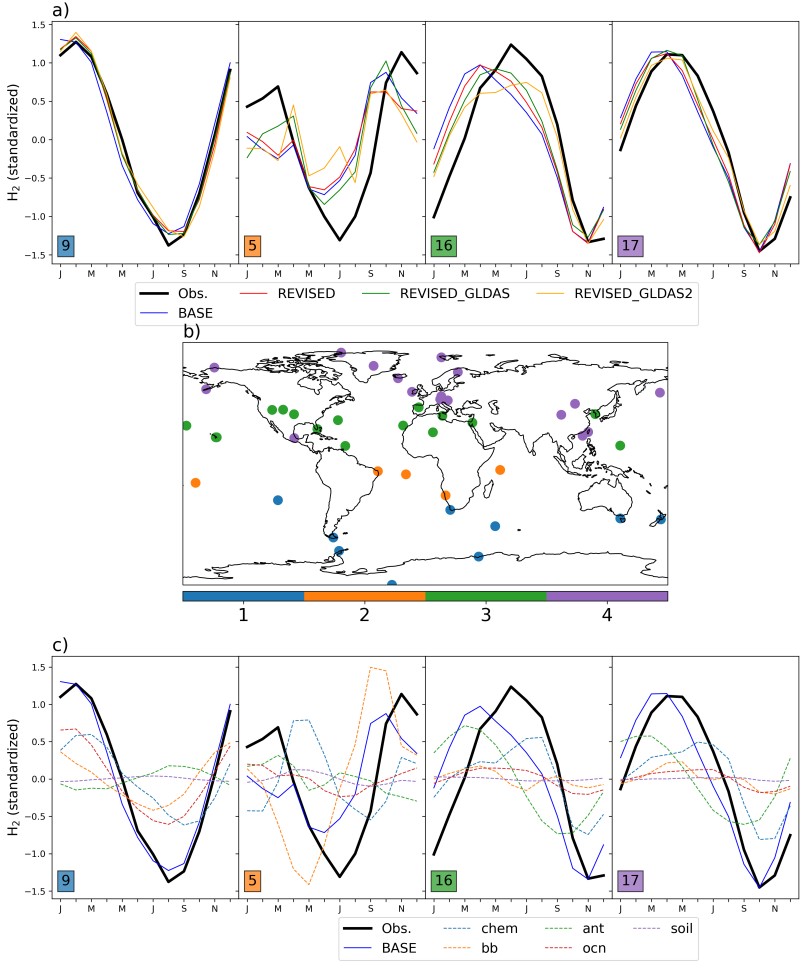

**Figure 3.** Monthly standardized $H_2$ concentration for each cluster (a). The number of sites in each cluster is indicated by insets. The sites included in each cluster are shown in panel (b). The variation of source-tagged $H_2$ tracers in each cluster is shown in panel (c). Source-tagged $H_2$ tracers are normalized using the standard deviation of simulated $H_2$.

## 4   Discussion

The BASE simulation was tuned against seasonal mean ground-based $H_2$ mole fraction reported by NOAA, CSIRO and AGAGE over the 1995-2005 period (Paulot et al., 2021). As detailed in Pétron et al (in preparation), the X1996 calibration scale used for NOAA observations for the 1995-2005 period induced not only a bias but also a drift in NOAA $H_2$ observations. The model evaluation against the more recent and recalibrated NOAA dataset highlights significant biases in the simulated mean concentration, trend, and seasonality of $H_2$ in the BASE configuration (section 3). Here, we evaluate the constraints that the recalibrated NOAA observations imply for $H_2$ emission and soil uptake.





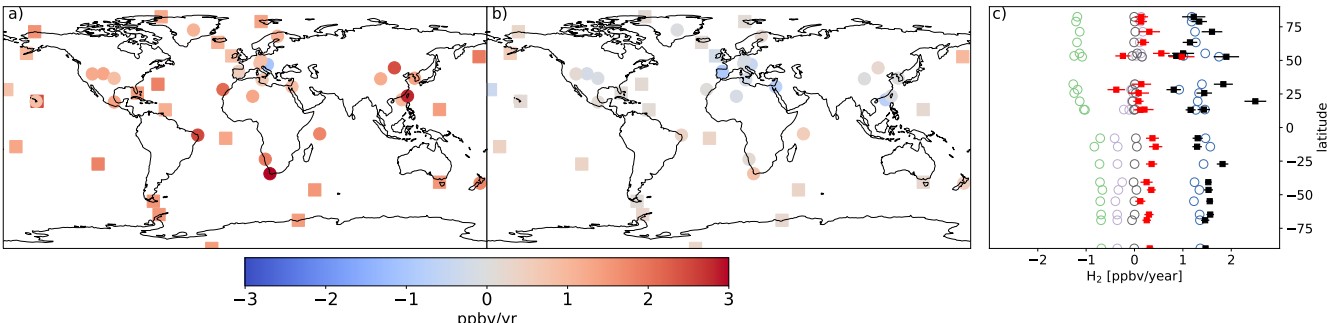

**Figure 4.** Trend in $H_2$ concentrations in observations (a) and in the BASE simulation (b) over the 2010–2019 period. Panel (c) shows the observed (black) and simulated (red) trend in $H_2$ at background sites (squares) as well as the trend in tagged $H_2$ tracers associated with anthropogenic sources (green), biomass burning (purple), ocean+soil sources (black), and photochemical production (blue). The error bars show one standard deviation for the estimated observed and simulated trends.

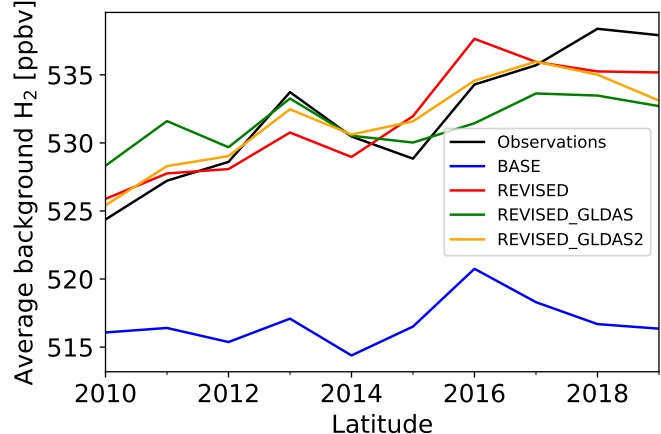

**Figure 5.** Mean observed and simulated $H_2$ at background sites (see Fig. 2 for locations)

## 4.1 Emissions

Following Ghosh et al. (2015), the changes in the $H_2$ source ($\Delta S(H_2)$) needed to reduce the model bias ($\Delta H_2(sfc)$) can be estimated as:

$$\Delta S(H_2) = K_1 \frac{d(\Delta H_2(sfc))}{dt} + K_2 \Delta H_2(sfc) \tag{1}$$

where $K_1$ is the ratio of the $H_2$ burden to the surface concentration of $H_2$, $K_2$ is the ratio of the loss of hydrogen to the surface concentration of $H_2$, and $\Delta H_2(sfc)$ is the difference between observed and simulated $H_2$ at background sites (Fig. 5). Equation 1 yields an estimated missing source of $H_2$ of $\simeq$ 2-2.5 Tg/yr circa 2010 and 3-4 Tg/yr circa 2019. The inferred increase in $H_2$





emissions over the 2010–2019 is of similar magnitude to the decline in anthropogenic emissions in our BASE simulation (Fig.
1) and we focus on this term in this section.

In the BASE simulation, $\simeq 80\%$ of $H_2$ emission originate from the transportation and residential sectors (Fig. 6a). Global
anthropogenic emissions are 1.4 Tg/yr lower in 2019 compared to 2010, with the largest decline from the transportation
(-1 Tg/yr) and industrial (-0.4 Tg/yr) sectors, respectively. Fig. 6b shows a revised anthropogenic inventory for $H_2$, which
is described in Appendix A1. The revised inventory incorporates a more detailed treatment of transportation and industrial
emissions. In particular, we include $H_2$ leakage from industrial production of $H_2$ for refining, ammonia, methanol and steel
production, assuming a time-invariant leakage rate of 2%, consistent with recent estimates (2.7% (Fan et al., 2022), 1.2%
(Arrigoni and Bravo Diaz, 2022)). We estimate that the increase in $H_2$ demand from these sectors ($+\simeq 18$ Tg/yr in 2019 relative
to 2010 (International Energy Agency, 2019)) has resulted in $\simeq 0.3$ Tg/yr more $H_2$ emissions over the 2010-2019 period. The
REVISED anthropogenic emissions are estimated to be 14.1 Tg/yr in 2010 and 13.5 Tg/yr in 2019, a lower decrease than
in the BASE configuration, which is consistent with the missing emissions inferred from equation 1. However, the updated
treatment of anthropogenic emissions does not explain the low bias in the simulated $H_2$ mixing ratio. Ehhalt and Rohrer (2009)
surveyed many "minor" sources of $H_2$, the combined magnitude of which could amount to 2 Tg/yr. For instance, we do not
include geological sources of $H_2$, the magnitude of which carries considerable uncertainty (0-30 Tg/yr (Zgonnik, 2020)). In
the REVISED simulation, we increase the $H_2$ soil source from 3 Tg/yr to 4.5 Tg/yr as described in Appendix A2. Clearly more
observational constraints are needed to develop a more robust $H_2$ emission inventory.

We find that the REVISED configuration exhibits reduced mean bias against observations for both the mean (Fig. 2) and the
trend (Figs 7 and 5). In contrast, the simulated North-South gradient (Fig. 2) and the $H_2$ seasonal cycle (Fig. 3) exhibit little
sensitivity to the change in emissions.

## 4.2  Deposition

In the previous subsection, we explored how changes in $H_2$ sources impact the model bias. In this section, we focus on the
representation of the soil removal of $H_2$, the largest sink of atmospheric $H_2$.

The soil removal of $H_2$ is controlled by the activity of high-affinity hydrogen oxidizing bacteria (HA-HOB, Constant et al.
(2010)). While considerable progress has been made in the last decade to characterize these organisms (Greening et al., 2015),
their representation in global models remains simplistic (Paulot et al., 2021). $H_2$ uptake has been shown to be very sensitive to
soil moisture (Smith-Downey et al., 2006). This reflects the competition between the biological uptake of $H_2$, which tends to
increase with soil moisture and the diffusion of $H_2$, which decreases with soil moisture (Bertagni et al., 2021). Furthermore, $H_2$
uptake has been shown to be inhibited when soil moisture is very low (Smith-Downey et al., 2006; Ehhalt and Rohrer, 2011).

To quantify possible changes in the soil removal of $H_2$ over the 2010-2019 period, we perform additional simulations using
3-hourly soil moisture and soil temperature from the NASA Global Land Data Assimilation System (Rodell et al., 2004) as
described in Appendix B. As in the BASE configuration, the deposition parameterization follows (Ehhalt and Rohrer, 2013)
except for the parameterization of the soil moisture response of HA-HOB activity, which follows Bertagni et al. (2021). The
parameterization of Bertagni et al. (2021) relates the minimum moisture threshold required for $H_2$ uptake by HA-HOB to soil





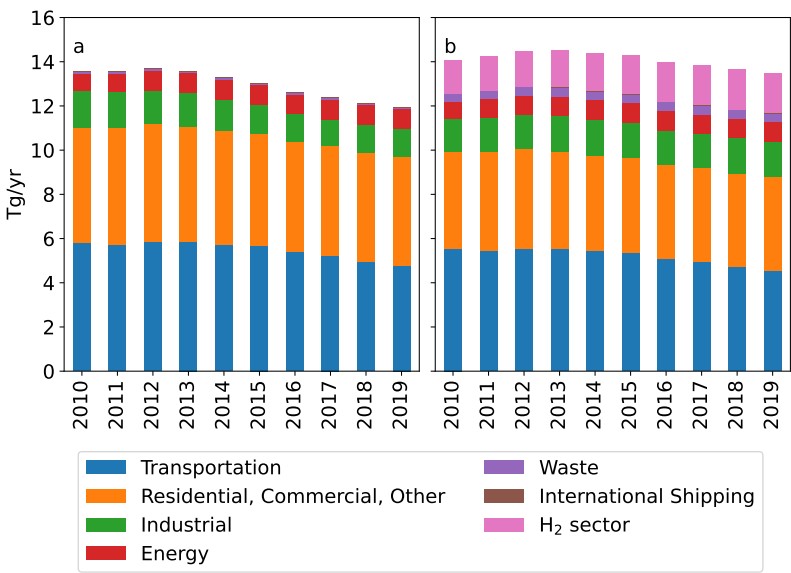

**Figure 6.** Sectorial $H_2$ anthropogenic emissions in the BASE (a) and REVISED (b) configuration

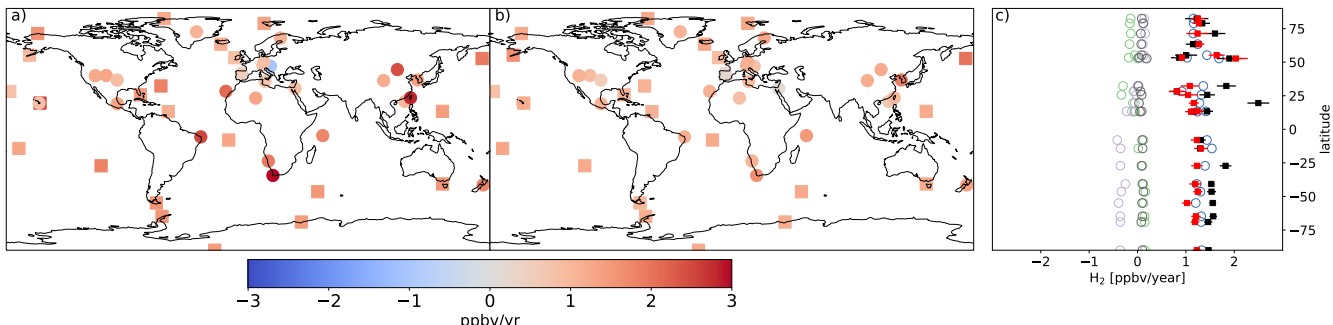

**Figure 7.** Same as Fig. 4 but for the REVISED configuration





hydrological properties, which facilitates its incorporation in global models. Here, we assume that $H_2$ uptake is inhibited when the soil matrix potential is lower than $\Psi_{ws} = -3000 kPa$ (Bertagni et al., 2021). This configuration, including the REVISED

emissions, is referred to as REVISED_GLDAS hereafter (Table 1).

Fig. 8 shows that the resulting $v_d(H_2)$ exhibits a different meridional distribution relative to the BASE configuration with faster removal in the subtropics and northern high latitudes but slower removal in the tropics. This reflects more efficient removal of hydrogen in arid regions and slower removal in tropical savanna than in the BASE configuration. Fig. 8b further shows that $v_d(H_2)$ in REVISED_GLDAS increases from 2009 to 2019 in the Northern mid latitudes. This increase reflects

drier and warmer conditions in Europe, the Western US as well as parts of Siberia, which result in faster biological uptake rates and promote $H_2$ diffusivity (Fig. A3). This mechanism may explain the reported 1.2%/yr increase in $H_2$ deposition velocity at Mace Head from 1994 to 2020 (Derwent et al., 2021). In contrast, drier conditions in Australia are projected to trigger biotic limitations, which results in a large decrease in $H_2$ deposition velocity in the Southern mid latitudes.

Changes to the spatial distribution of $v_d(H_2)$ and the increase in $H_2$ removal in the Northern mid latitudes (Fig. 8b) in

REVISED_GLDAS result in a larger pole-to-pole difference in surface $H_2$ (Fig. 2) and a reduction in the simulated trend (Fig. 10) in the Northern mid to high latitudes. Both of these changes tend to degrade the model performance relative to the REVISED configuration. In contrast, the REVISED_GLDAS configuration better captures the timing of the $H_2$ maximum in the northern hemisphere (clusters 3 and 4, Fig. 3).

Experimental studies have shown that HA-HOB are present in very arid environments and strongly stimulated by wetting

(Jordaan et al., 2020). However, the soil moisture required for $H_2$ uptake remains poorly constrained. We thus conduct a range of sensitivity simulations to systematically test the dependence of $v_d(H_2)$ to $\Psi_{ws}$ (see Appendix B). Fig. 9a shows that a lower soil moisture threshold for HA-HOB activation (i.e., a lower $\Psi_{ws}$) favors $H_2$ removal in the Northern hemisphere relative to the Southern hemisphere (Fig. 9a) and results in a larger increase in $v_d(H_2)$ over the 2009–2019 period (Fig. 9b), especially in the Southern hemisphere (Fig. 9c). This suggests that a lower $\Psi_{ws}$ would tend to worsen the model performance (given the

REVISED emissions).

Previous studies have also shown that $H_2$ uptake by HA-HOB can be reduced by litter (Smith-Downey et al., 2008; Ehhalt and Rohrer, 2009), which acts as a barrier for the diffusion of $H_2$ to active sites. We find that such a barrier tends to increase the gradient in $v_d(H_2)$ between Northern and Southern hemisphere (Fig. 9a) and to reduce (or even reverse) the increase in $v_d(H_2)$ (Fig. 9b).

It is notable that no configuration results in little change in $v_d(H_2)$ without producing large and increasing gradients between Northern and Southern hemisphere. As a result, our model cannot reproduce trends, meridional gradient, and seasonality together given our best estimate of $H_2$ emissions (REVISED configuration). This is illustrated by the REVISED_GLDAS2 configuration in which we use a lower moisture threshold ($\Psi_{ws}$=-10000 kPa) and account for both the impact of litter and canopy on $H_2$ soil uptake (Litter scale=1). This configuration is found to improve the simulated trend relative to the REVISED_GLDAS

(not shown) and the seasonality relative to the REVISED configuration (Fig. 3) but results in a larger overestimate of the South/North meridional gradient than the REVISED_GLDAS configuration (Fig. 2).

This highlights the need for a more detailed representation of the factors that modulate HA-HOB (Khdhiri et al., 2015).



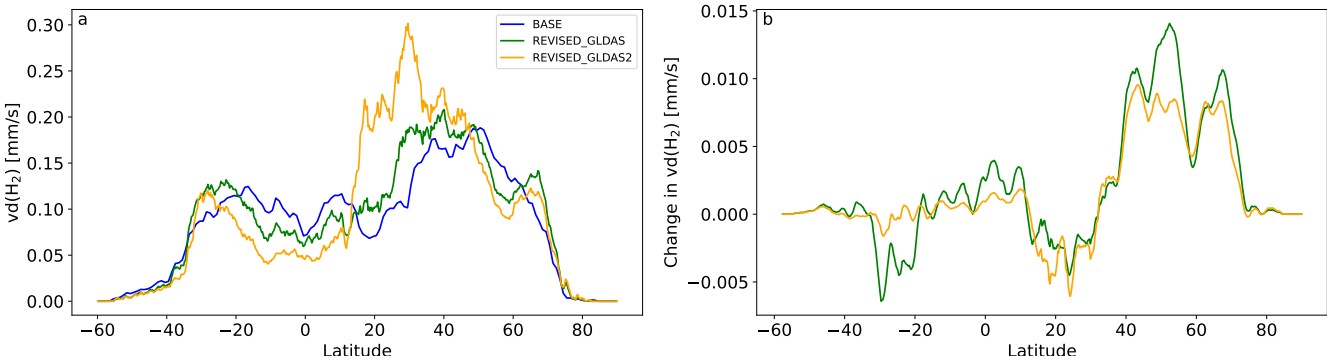

**Figure 8.** Meridional distribution of $v_d(H_2)$ in the BASE, REVISED_GLDAS, and REVISED_GLDAS2 simulations (a) and (b) simulated change in $v_d(H_2)$ between (2015–2019) and (2009–2013)

## 5  Conclusions

The recently released $H_2$ dry air mole fraction measurements from the NOAA Global Cooperative Air Sampling Network
expand the spatial coverage of the WMO Global Atmospheric Watch observations. This offers the opportunity to assess the
representation of the $H_2$ atmospheric budget in the state-of-the-art GFDL-AM4.1 global atmospheric chemistry climate model.
Observations show that $H_2$ has increased on average by 1 to 2 ppbv/year over the 2010-2019 period. This can be explained by
the increase in photochemically-produced $H_2$ (mostly from $CH_4$) provided direct anthropogenic $H_2$ emissions have remained
stable during this time period. We hypothesize that this stability reflects the compensation between declining emissions asso-
ciated with fossil fuel combustion (mostly from the transport sector) and increasing emissions associated with $H_2$-producing
facilities (primarily for ammonia ($NH_3$) and methanol production as well as refineries). This is notable as $H_2$ release from $H_2$
production facilities is poorly understood yet critical to assess the climate benefits of $H_2$ (Hauglustaine et al., 2022; Bertagni
et al., 2022).

We show that the observed trend, seasonality, and meridional gradient of $H_2$ provide complementary constraints on the
global $H_2$ biogeochemical cycle. We find that our model fails to capture all three constraints together, which likely reflects
fundamental gaps in our representation of the soil removal of $H_2$ by microorganisms (HA-HOB). In particular, we find that
the sign of the simulated global trend in soil $H_2$ removal over the 2010–2019 period is sensitive to the soil moisture threshold
below which the activity of HA-HOB is suppressed.

This highlights the need for coordinated field and laboratory data collection efforts to help improve models of the distribution
and activity of HA-HOB in global models (American Academy of Microbiology, 2023). Such efforts are currently hindered by
the lack of sensors that offer higher time resolution and maintain good sensitivity and stable response. Such efforts are critical
to quantify the response of atmospheric $H_2$ to increasing anthropogenic $H_2$ usage as well as hydrological changes associated
with climate change (Jansson and Hofmockel, 2019; Huang et al., 2015).



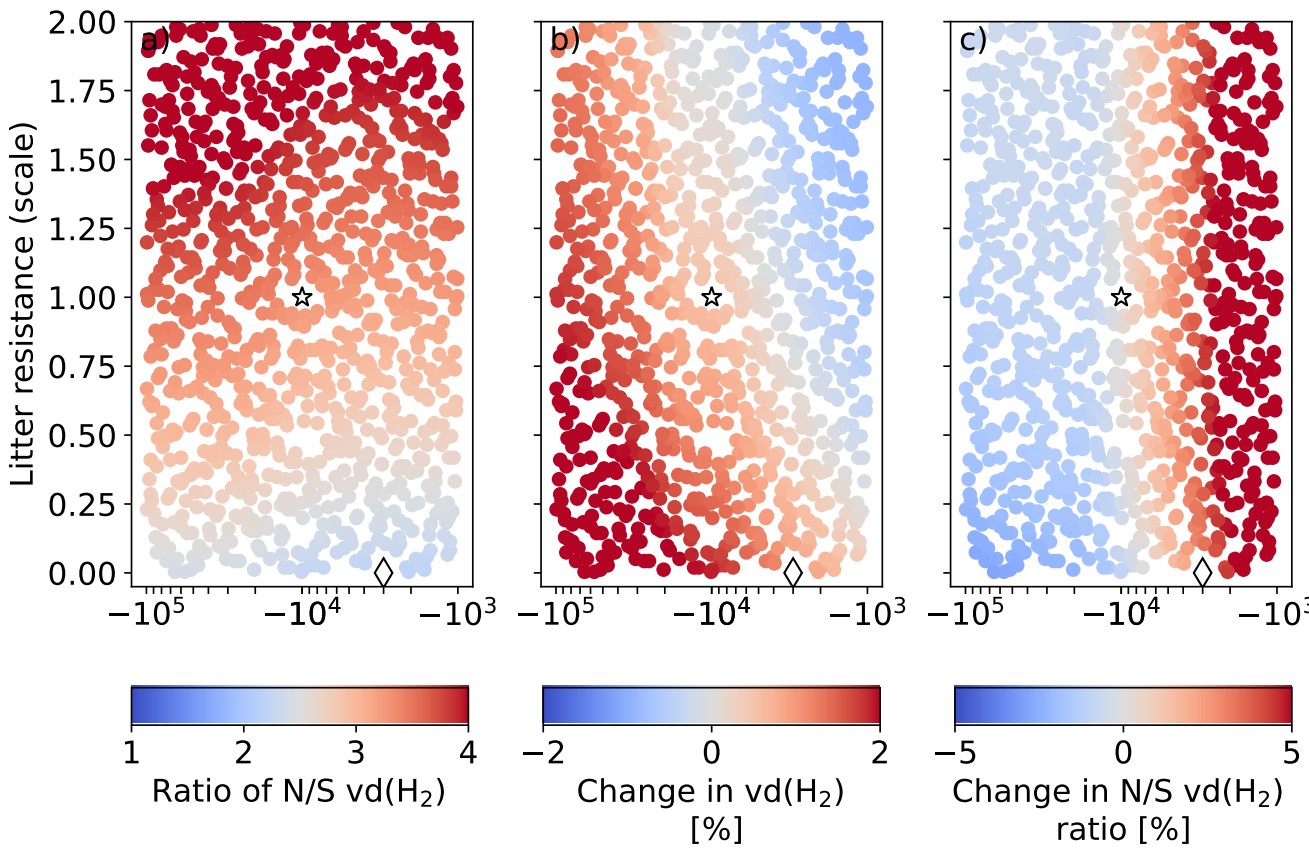

**Figure 9.** Simulated sensitivity of $v_d(H_2)$ to $\Psi_{ws}$ and the strength of the litter diffusive barrier. Panels a, b and c show the response of the North/South ratio of $v_d(H_2)$, the difference in $v_d(H_2)$ in (2015–2019) relative to (2009-2013), and the difference in the N/S $v_d(H_2)$ gradient in (2015–2019) relative to (2009-2013). The REVISED_GLDAS configuration uses $\Psi_{ws} = -3000$ kPa and no litter resistance (diamond). The REVISED_GLDAS2 uses $\Psi_{ws} = -10000$ kPa and a litter resistance scale of 1 (star).

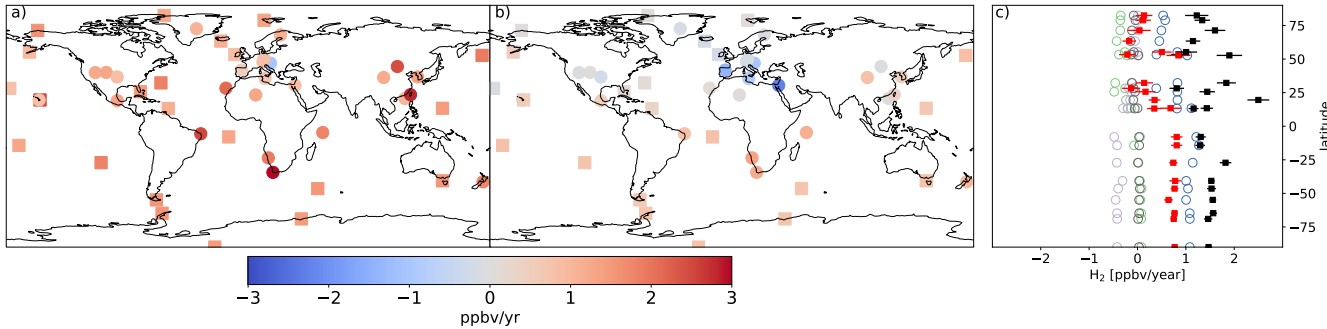

**Figure 10.** Same as Fig. 4 but for the REVISED_GLDAS configuration



*Code and data availability.* The code for the GFDL ESM4.1 model is available at https://zenodo.org/record/3836405. NOAA Global Cooperative Network Flask Air $H_2$ (Pétron et al., 2023) can be downloaded at https://doi.org/10.15138/WP0W-EZ08.

## Appendix A: Revised emission inventory

The $H_2$ budget in the REVISED experiment is summarized in Fig. A1. Anthropogenic and natural emissions are described below.

### A1 Anthropogenic emissions

**Table A1.** Sector-based molar $H_2$ to CO emission ratio

|  |  | BASE[a] | REVISED |
|---|---|---|---|
| Industrial |  | 0.2 | 0.2 |
| Residential |  |  |  |
|  | Biofuel | 0.3 | 0.31 [b] |
|  | Other | 0.3 | 0 [c] |
| Transportation |  |  |  |
|  | Gasoline-powered vehicles (up to EURO3) | 0.5 | 0.5 [d] |
|  | Gasoline-powered vehicles (EURO4 and above) | 0.5 | 1 [d] |
|  | Diesel-powered vehicle | 0.5 | 0.0021 [d] |
|  | CNG-powered vehicle | 0.5 | 0.04 [d] |
| Waste |  | 0.07 | 0.32 [b] |

[a] Paulot et al. (2021) [b] Andreae (2019) [c] Vollmer et al. (2012) [d] Bond et al. (2010, 2011)

In the BASE simulation, anthropogenic emissions are assumed to solely originate from combustion processes and calculated using time-invariant and source-specific $H_2$ to CO emission ratios (Table A1) that reflect the water–gas shift reaction.

The REVISED emission inventory incorporates a more detailed treatment of $H_2$ emission factors. In particular, we account for the difference between gasoline- and diesel-powered vehicles and for the increase in the $H_2$ to CO emission ratio associated with three-way catalytic converters (Bond et al., 2010, 2011). $H_2$ vehicular emissions are estimated using $H_2$:CO emissions ratio (Table A1) and ECLIPSEv6 CO region- and vehicle-type specific emissions (Klimont et al., 2017). These changes result in a model decrease in transportation emissions in 2010 (5.5 Tg/yr vs 5.8 Tg/yr).The REVISED emission ratio for biofuel and waste are from Andreae (2019). Following Vollmer et al. (2012), we assume that other residential emissions of CO (e.g., oil and gas stoves) do not produce $H_2$.

The industrial emission ratio is not modified between the BASE and REVISED emissions inventories. However, in the REVISED inventory, we use the Emissions Database for Global Atmospheric Research (EDGAR) v6.1 industrial CO emissions





instead of CEDS to estimate industrial $H_2$ emissions. These inventories exhibit different trends for CO (+8.7 Tg/yr for EDGAR and -30.7 TgTg/yr for CEDS in 2018 relative to 2010), which translate to different trends in $H_2$ emissions (+0.1 Tg/yr and -0.4 TgTg/yr, respectively). We select the EDGAR inventory as we identified the decrease in industrial $H_2$ as one of the main drivers for the decline in anthropogenic emission in the BASE inventory.

The REVISED inventory also includes a non-combustion source of $H_2$ associated with $H_2$ industrial production (primarily for $NH_3$ production and refining (International Energy Agency, 2019)). Using a 2% release rate (Bond et al., 2010) yields an estimated source of 1.5 Tg/yr in 2010 and 1.8 Tg/yr in 2019. The increase in $H_2$ thus contributes the largest increase in $H_2$ emissions over the 2010 to 2019, which highlights the need to better quantify $H_2$ leakage throughout the $H_2$ supply chain.

## A2    Natural emissions

The magnitude of natural emissions in the BASE configuration (9 Tg/yr) is similar to that of anthropogenic emissions ($\simeq$ 13 Tg/yr) with considerable uncertainties (Ehhalt and Rohrer, 2009). In the BASE configuration, soil and ocean emissions are 3 and 6 Tg/yr respectively (Ehhalt and Rohrer, 2009) and are distributed based on the soil and marine CO emission patterns in the Precursors of Ozone and their Effects in the Troposphere inventory (Granier et al., 2005).

     In the REVISED inventory, marine $H_2$ emissions are calculated interactively (Johnson, 2010; Paulot et al., 2021) from the
simulated distribution of surface seawater CO (Conte et al., 2019), scaled to produce a net flux of 6 Tg/yr. We use CO as a proxy for biological activity following Pieterse et al. (2011). Relative to the BASE inventory, the REVISED inventory exhibits higher emissions in the tropics and lower emissions in the Southern ocean, which reflects changes in the solubility of $H_2$ (Fig. A2a).

     The soil source of $H_2$ is distributed following the simulated land biological nitrogen fixation from the MIROC-ES2L Earth
system model (Hajima et al., 2020). The soil $H_2$ flux is set to 4.5 Tg/yr, which is at the high end of previous estimates (Ehhalt and Rohrer, 2009). MIROCA-ES2L explicitly accounts for biological nitrogen fixation by crops. This results in much larger $H_2$ emissions in the Northern mid latitudes relative to the BASE soil emissions.

     Biomass burning emissions are kept unchanged from Paulot et al. (2021). However, we note that using the emission factors of Andreae (2019) would reduce $H_2$ emissions from 8.3 to 6.1 Tg/yr over the 2010–2019 period.

## 295   Appendix B: Deposition sensitivity

The deposition velocity of $H_2$ can be expressed as

$$\frac{1}{v_d(H_2)} = \frac{1}{g_i} + \frac{1}{g_s} \tag{B1}$$

where $g_i$ and $g_s$ represent the $H_2$ conductance through barriers that reduce the transport of $H_2$ to active sites (e.g., canopy, litter, ...) and in the soil.

The conductance in the soil is expressed after Ehhalt and Rohrer (2013) as

$$g_s = \sqrt{k_m \, hT f \, D_s} \tag{B2}$$



where $hT$ and $f$ are the sensitivity of $H_2$ biological uptake to temperature and soil moisture, respectively, $D_s$ is the moisture-dependent diffusivity of $H_2$ in the soil, and $k_m$ represents the maximum uptake rate of $H_2$. All moisture dependencies are evaluated after Bertagni et al. (2021). Namely, $f$ is expressed as

$$f(s) = \frac{1}{N}(s - s_{ws})^{\beta_1}(1 - s_{ws})^{\beta_2} \tag{B3}$$

where $s_{ws}$ is the threshold below which $H_2$ consumption is inhibited. $s_{ws}$ can be estimated as:

$$s_{ws} = \left(\frac{\tilde{\Psi}}{\Psi_{ws}}\right)^{\frac{1}{b}} \tag{B4}$$

where the $\tilde{\Psi}$ and $b$ constants can be determined experimentally (Bertagni et al., 2021) and $\Psi_{ws}$ is the soil matrix potential below which bacterial uptake is inhibited. Given $s_{ws}$, $\beta_1$ and $\beta_2$ can be estimated based on observational constraints (Bertagni
et al., 2021).

For $g_i$, we account for the impact of canopy and above-ground litter. For the canopy, we assume a time-invariant conductance based on the vegetation type (Makar et al., 2018). The litter conductance is estimated assuming a litter porosity of 0.62 (Wang et al., 2019). The litter depth is estimated based on the simulated above ground carbon from the IPSL INCA model historical simulation (Boucher et al., 2021) assuming a density of 0.03 $g/cm^3$ (Chojnacky et al., 2009).

We carry sensitivity experiments in which the resistance due to litter and canopy conductance are scaled by a factor between 0 and 2 and $\Psi_{ws}$ takes values between $-10^5$ and $-10^3$ kPa (compared to -3000 kPa in REVISED_GLDAS). For each combination, $k_m$ is optimized to yield the same global $v_d(H_2)$ for year 2010. We find that the canopy resistance has little impact on the meridional gradient and trend and we focus our analysis on the litter resistance.

*Author contributions.* FP designed the research, developed, and analyzed the model simulations. GP and AC collected and processed $H_2$
observations from the NOAA network and provided guidance regarding their interpretation. MB developed the soil moisture parameterization of HA-HOB used in the REVISED_GLDAS and REVISED_GLDAS2 configurations. All authors contributed to the drafting of the manuscript.

*Competing interests.* None

*Acknowledgements.* We thank Vaishali Naik for her help with generating model-ready $H_2$ emissions. We thank Larry Horowitz, Vaishali
Naik, Amilcare Porporato, and Xinning Zhang for their helpful comments on the manuscript. This research was supported in part by NOAA cooperative agreements NA17OAR4320101 and NA22OAR4320151 and by the U.S. Department of Energy, Office of Energy Efficiency and Renewable Energy (EERE), specifically the Hydrogen and Fuel Cell Technologies Office. The views expressed herein do not necessarily represent the views of the U.S. Department of Energy or the United States Government.



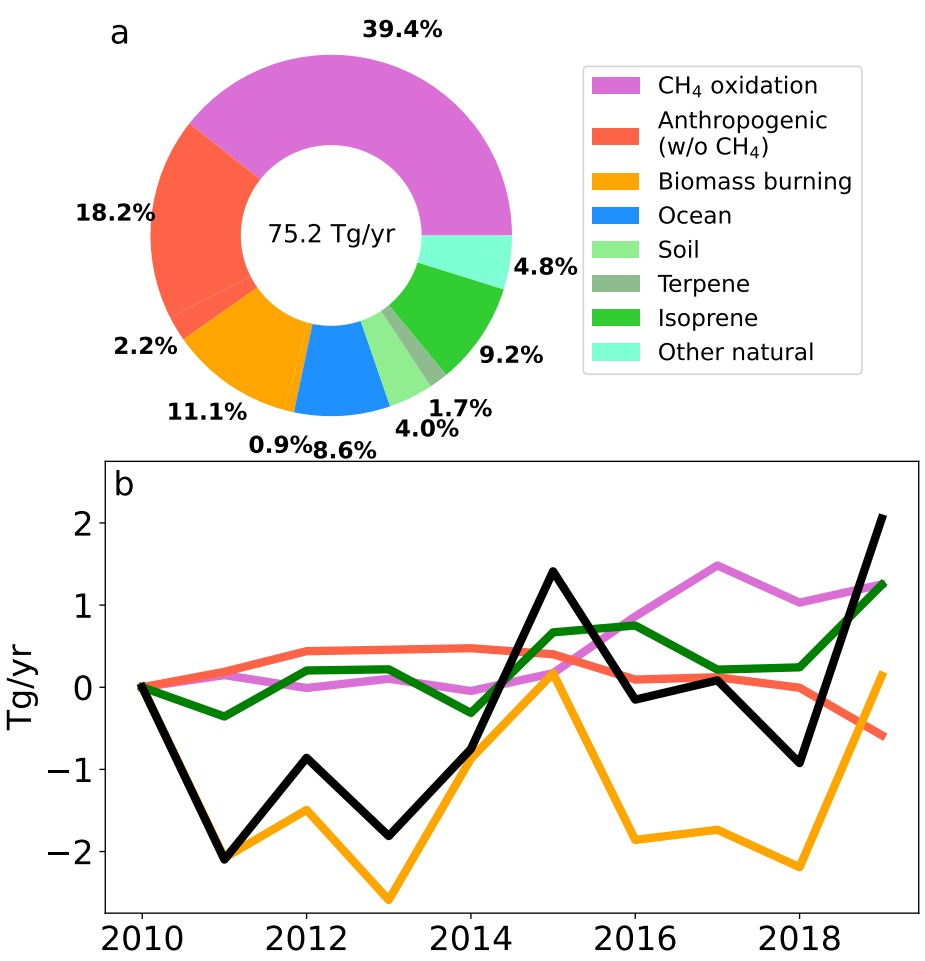

**Figure A1.** Same as Fig. 1 for the REVISED experiment.

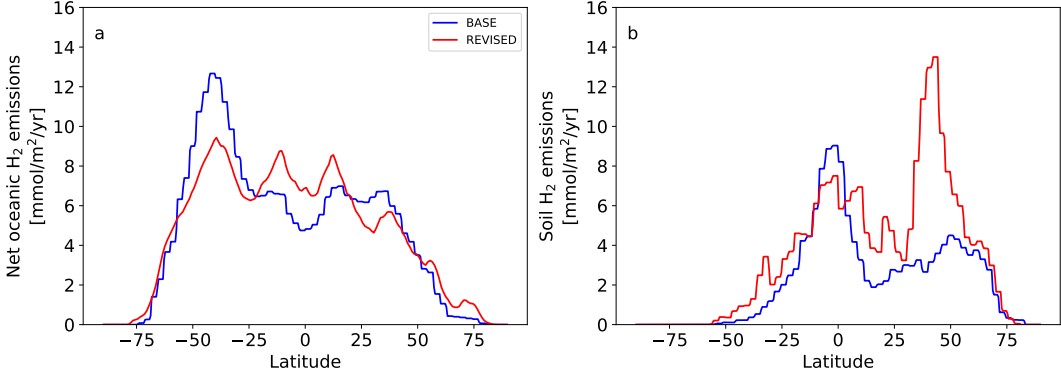

**Figure A2.** Marine and soil $H_2$ emissions in the BASE and REVISED emission inventories.



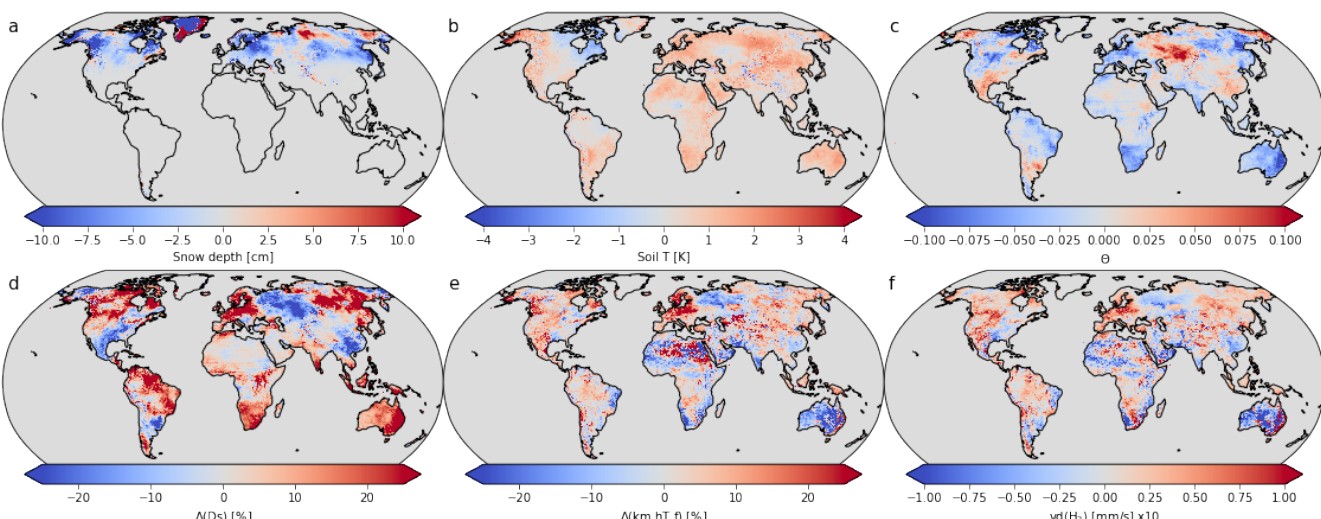

**Figure A3.** Changes in snow depth (a), soil temperature (b), soil moisture (as a fraction of pores (c)) and their impact on $H_2$ soil diffusivity (d), $H_2$ bacterial uptake, (e) and $H_2$ deposition velocity (REVISED_GLDAS, panel f) between years (2015–2019) and years (2009–2012).

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
