# Peer review of "Reanalysis of NOAA H2 observations: implications for the H2 budget"

_EGUsphere, 2023_

## Author Comment (AC1)

We thank both reviewers for their careful review and constructive comments. Please find our replies below (red) as well as changes made to the manuscript (blue)

**Reviewer 1**

**This study uses a recently completed recalibration of atmospheric hydrogen measurements from 2010 to 2019 to evaluate how well a state-of-the-art chemistry-climate model – that has been used to evaluate hydrogen's climate effects - represents hydrogen concentrations. Initially, the authors do not consider hydrogen emissions from the hydrogen industry, and then ultimately show that when hydrogen leakage is considered, the model does a much better job capturing the temporal trend. However, the deposition of hydrogen via the soil sink remains highly uncertain, and different model configurations to be more comprehensive are each unable to reproduce trends, spatial gradients, and seasonality. Overall, I find this paper important, timely, technically thorough and sound, and well-written. As plans to scale up hydrogen advance, it is important that climate modelers are able to perform experiments to assess the resulting climate implications. However, there remain major gaps in our ability to model hydrogen systems. This paper advances our understanding of modeling hydrogen in sophisticated models, which will be increasingly important. I have one major comment regarding how the manuscript is organized, and several minor comments.**

We thank the reviewer for his/her detailed review and suggestions. We address each specific comment below.

**Major comments**

**I recommend reorganizing some of the content in the paper; some of the content feels out of place. Specifically, the introduction is really short and I find it does not adequately set up the information needed to understand the methods and results. For example, a key element to understanding hydrogen concentrations is hydrogen sources and sinks. I would expect the state of the science of hydrogen's sources and sinks to be discussed in the introduction, but they are not discussed in detail until the methods, in which they are then mixed in with the references of where the data was taken from. I think it would strengthen the paper if discussion of the sources and sinks was first brought up in the introduction, and then details of model inputs and sourcing would follow later in the methods.**

Thank you for this recommendation. We have revised the introduction to include a more comprehensive discussion of the uncertainties in the present-day budget of $H_2$

*Significant uncertainties regarding the overall budget of $H_2$ remain. $H_2$ sources include both emissions and photochemical production from the oxidation of volatile organic compounds (VOCs). Estimates for the overall source of atmospheric $H_2$ range from $\simeq$ 70 to 110 Tg/yr, a large spread primarily associated with the magnitude of the $H_2$ photochemical sources (Ehhalt and Rohrer, 2009). Recent work also argues that current estimates of $H_2$ sources need to be revised upward to account for geologic $H_2$ seepage (Zgonnik, 2020). These uncertainties in the nature and magnitude of $H_2$ sources have proved challenging to reduce in part because of commensurate uncertainties in $H_2$ sinks. The atmospheric oxidation of $H_2$ by OH is well understood but is estimated to account for less than one third of the overall sink (Ehhalt and Rohrer, 2009; Paulot et al., 2021). The most important removal pathway is the consumption of $H_2$ by high-affinity hydrogen oxidizing bacteria (HA-HOB), a class of bacteria that have been identified in many different soils (Constant et al., 2008; Greening et al., 2015; Bay et al., 2021; Greening and Grinter, 2022). Several parameterizations of the $H_2$ soil sink have been developed (Ehhalt and Rohrer, 2013; Price et al., 2007; Smith-Downey et al., 2006; Bertagni et al., 2021) that aim at capturing the observed sensitivity of $H_2$ soil removal to soil temperature, soil moisture and ecosystem/soil type (Ehhalt and Rohrer, 2009). However, observational constraints on $H_2$ soil removal remain very limited (Meredith et al., 2016) and this process remains challenging to represent in global models (Yashiro et al., 2011; Paulot et al., 2021).*

**Further, I find it a bit odd that the sensitivity methods and results are in the discussion section. I understand that their purpose likely came about from the model results of the base case, and therefore it makes sense chronologically based on the research timeline that they are discussed thereafter. However, as a reader, I want to see all of the methods etc. together in the same section, results in the same section, and discussion in the same section. Right now I had to go back and forth between sections to fully comprehend the experiments and analysis and the takeaways. Therefore, I suggest moving the sensitivity methodology into the methods section, the sensitivity results into the results section, and then leaving the discussion for interpretation and implications of the combined set of results.**

Thank you for this recommendation. We have expanded the method section to include a description of the emission inventory and deposition parameterizations that are used in both the BASE and different sensitivity simulations. We have also reorganized the Results and Discussion sections into two sections: BASE model evaluation and Sensitivity simulations.

**Minor comments**

1. **Abstract: "Robust assessment of the climate implications of increased H2 usage in the global economy is partly hindered." Just a not regarding the use of the word robust – some may misinterpret this to mean that we don't have robust assessment of hydrogen's warming effects, whereas the authors' own recent study (Sand et al. 2023) suggested otherwise.**

We have replaced this sentence by:

*However, significant gaps in our understanding of the atmospheric* $H_2$ *budget limit our ability to predict the impacts of greater* $H_2$ *usage.*

2. "but the increased demand for CH4 may offset much of the expected climate benefit of increased H2 usage" Would be helpful to non-experts if it is briefly explained with the demand for CH4 could offset expected climate benefits (i.e. CH4 emissions)

We have revised the text as follows:

*However, methane leakage throughout the supply chain could offset much of the expected climate benefits of increased* $H_2$ *usage (Howarth and Jacobson, 2021; Ocko and Hamburg, 2022; Bertagni et al., 2022; Hauglustaine et al., 2022).*

3. **2-24: "evidence of high concentrations of H2 in many different geologic environments" – my read of the literature is that this sentence overstates the evidence, so I would temper a bit.**

We agree with the reviewer and have revised the statement accordingly. We have also added a reference to the recent studies of Milkov (2022) and Lefeuvre et al. (2021)

*Furthermore, evidence of significant concentrations of* $H_2$ *in surface and subsurface natural gases (Zgonnik, 2020; Milkov, 2022; Lefeuvre et al., 2021) have spurred interest in the potential of naturally occurring* $H_2$ *as a new primary energy source (Prinzhofer et al., 2018; Lapi et al., 2022).*

4. **2-28: Update to reflect Sand et al. 2023, suggest acknowledging that H2 warming effects are short-lived and adding the value for GWP20 given that challenges of using GWP100 for short-lived warming effects.**

We have revised the text as follows:

*Sand et al. (2023) recently calculated that* $H_2$ *has a global warming potential of* $\simeq 11.6 \pm 2.8$ *and* $37.3 \pm 15.1$ *for a 100 and 20-year time horizon, respectively.*

5. **"60+ globally distributed sites." Are they the locations in Fig 2? Can that be referenced here? Or perhaps a clearer map of the locations is needed beyond just a comparison to model data.**

No, we only use a subset (47 stations) of these stations. We have added a link to the NOAA CCGG program `https://www.gml.noaa.gov/dv/site/?program=ccgg`

We have also added a Figure showing the location and name of all the stations used in this study (Fig. S1)

[Figure]

Figure S1: Location of the ground surface stations used in this study. 1.ALT 2.ASC 3.ASK 4.BHD 5.BMW 6.BRW 7.CBA 8.CGO 9.CIB 10.CPT 11.CRZ 12.DSI 13.EIC 14.GMI 15.HPB 16.HUN 17.ICE 18.IZO 19.KEY 20.KUM 21.LLN 22.LMP 23.MEX 24.MHD 25.MID 26.MLO 27.NAT 28.NMB 29.NWR 30.OXK 31.PAL 32.PSA 33.RPB 34.SEY 35.SGP 36.SHM 37.SMO 38.SPO 39.SUM 40.SYO 41.TAP 42.USH 43.UTA 44.UUM 45.WIS 46.WLG 47.ZEP. Further information regarding each station can be found at Global Monitoring Laboratory (2023)

[Figure]

Figure 1: Global source of $H_2$ (panel a). Panel b shows the changes in the magnitude of $H_2$ sources over the 2010–2019 period. For clarity, the green line denotes the combined change in $H_2$ emissions and photochemical production from natural sources (marine and soil emissions + BVOCs photooxidation).

6. **"3-56: Reference still in prep?"**

   The dataset is publicly available (see data statement) and a manuscript (Pétron et al., submitted)) describing the calibration procedure is now under review in AMT

7. **"3-60/63: Again, a map showing this distribution and coverage would be useful.**

   See reply to comment 5

8. **Figure 1 – I think this figure could be greatly improved. These are prescribed emissions and not model results, or combo of both? The pie slices are not in the order of the legend, nor in the order of the magnitude, making it hard to follow. Why do some values have 1 significant figure and some have 3? Pie charts are also not the best for visualizing data with this many slices, a bar chart would be much clearer in terms of share of each source. Would also make clearer distinguishing the photochemical vs. direct sources (because those can be grouped next to one another), and you could add in the components that are anthropogenic via stacked bars (such as what fraction of methane/biomass burning/etc. are anthropogenic). Also what does "anthropogenic w/o CH4" mean?**

   We have revised Fig. 1 following the reviewer's suggestions. For $H_2$ emissions, the contribution of the different sectors is indeed from the inventory. For the photochemical sources of $H_2$, the production is estimated through a set of sensitivity simulations. This is now described in the Method section (see reply to detailed comment 2 from reviewer 2). Note that we corrected an overestimate in the estimated yield of $H_2$ from $CH_4$ and BVOCs reported in the preprint (see reply to Detailed comments 2 and 5 from reviewer 2)

**Reviewer 2**

This paper addresses an interesting issue related to H2, i.e., processes influencing the removal of H2 from Earth's atmosphere. The aim of the paper is good. However, I have some major issues noted below

   We thank the reviewer for his/her careful review. We have made significant changes to the manuscript in response to the reviewer's comments as detailed below.

**Major comments**

1. **The data used for the analysis here is not published. It includes recalibration of the older data. Until we have that data scrutinized and published, I do not see how this paper analyzing that unpublished data (from a different set of authors) can be published. Bottom line: How can I**

trust this analysis when I do not know if I trust the observational data used here? As the authors note, some observational data were selected and some were excluded. Maybe this analysis will hold up when the observational data is published, but I cannot take it for granted.

The data are publicly available at `https://doi.org/10.15138/WP0W-EZ08` as indicated in the data availability statement. Gabrielle Pétron and Andrew M. Crotwell, second and third authors, are responsible for the analysis of $H_2$ observations at NOAA GML. The manuscript detailing the calibration and validation of the NOAA observations is now under review for AMT.

2. **The paper is dense and very hard to follow. The analyses has so many moving parts that I cannot figure out how they homed in on soil uptake as the major issue that is influencing H2 trends.**

We have made the following changes to address the reviewers' comment (see also reply to comments 1 and 2 from reviewer 1). First, the introduction has been expanded to highlight gaps in our current understanding of $H_2$. We have also modified the structure of the paper, so that the different sensitivity simulations are described in the method section. Finally, we have added the following text to justify our focus on the deposition and emission terms:

*In this section, we describe additional model simulations that are designed to explore the impact of uncertainties in the representation of $H_2$ emission and deposition on the simulation of atmospheric $H_2$ (Table 1). We focus on $H_2$ emissions and deposition as their representations in models are largely derived from limited observational constraints (Derwent et al., 2023; Paulot et al., 2021).*

3. **It has no real estimate of uncertainties and discussion of the rather short period of 9 years for the reanalyzed data. The authors just assert some values without justification.**

We have added an estimate of the uncertainty in the observation section based on Pétron et al. (submitted). The error is on the order of 1–2 ppbv on an annual basis, which is much less than the observed trend over the 10 years that are considered here (15 ppb).

*NOAA reprocessed $H_2$ data since 2010 is consistent to within 1-2 ppbv on an annual basis for same air measurements with CSIRO and the MPI-BGC (Pétron et al., submitted).*

In the introduction (see reply to comment 2 from reviewer 1), we also present a detailed summary of the uncertainties in the present-day budget of $H_2$. The different treatments of anthrogenic activities and deposition that we consider in this study all fall well within the uncertainty in the deposition and emissions and thus can be understood as assessing the impact of errors in our understanding of H2 emissions. This is now explicitly stated in the preamble of the subsection entitled "Sensitivity simulations" (Method section)

*In this section, we describe additional model simulations that are designed to explore the impact of uncertainties in the representation of $H_2$ emission and deposition on the simulation of atmospheric $H_2$ (Table 1). We focus on $H_2$ emissions and deposition as their representations in models are largely derived from limited observational constraints (Derwent et al., 2023; Paulot et al., 2021).*

4. **I find the significant figures in this paper hard to swallow, especially when there are no real error estimates.**

Thank you. We agree with the reviewer that the number of significant figures didn't properly reflect the uncertainty. This has been adressed in the revised manuscript as exemplified by the revised budget figure (Fig. 1) and by the revised discussion of the different photochemical sources (see reply to detailed comment 2)

**Detailed comments**

1. **When did H2 from direct mining start and how much does it contribute to the emissions, especially over the period analyzed here?**
We are not aware of significant mining operations for $H_2$ and it is very unlikely that ongoing operations have a significant impact on the atmospheric $H_2$ budget. We have also clarified in the Method section that we do not include natural geological sources of $H_2$ in our current inventory

*The BASE emission inventory does not include possible geological sources of $H_2$.*

2. **How well do we know H2 production from CH4 oxidation? Even though CH4 is quite well mixed, what happens after its reaction with OH (that ultimately leads to H2) is very dependent on location and conditions.**
Our model does represent the key processes that control the fate of the methylperoxy radical (reaction with NO, HO2, and other peroxy radicals) and the chemistry of $CH_2O$ (deposition, photolysis, reaction with OH). We agree with the reviewer that different representation of VOC chemistry (e.g., CH2O quantum yield) and wet/dry losses will impact the simulated photochemical production of $H_2$. In the revised manuscript, we have

[Figure]

Figure S4: Simulated $CH_2O$ loss (a) and $H_2$ yield from $CH_2O$ and $CH_4$. The yields are estimated by dividing the annual column-integrated $H_2$ production associated with $CH_2O$ and $CH_4$ photooxidation by the column-integrated chemical and depositional loss of $CH_2O$ and by the phochemical loss of $CH_4$, respectively

included a comparison between the estimated $H_2$ yields from different VOCs derived from the GFDL model and from a box model (Grant et al., 2010). Furthermore, we have added a figure in the supporting materials that shows the simulated meridional variation in the production of $H_2$ from $CH_2O$ and from $CH_4$ (Fig. S4). We also emphasize the sensitivity of our results to the treatment of $CH_2O$ photolysis.

*To characterize the contribution of different VOC emissions to the photochemical production of $H_2$, we perform a set of sensitivity experiments in which we perturb the emission of a given VOC by 10% and quantify the response of $H_2$ production. For $CH_4$ oxidation, we directly track the different oxidation pathways that result in $H_2$ production. The molar yield of $H_2$ from $CH_4$, isoprene, methanol, and terpene are estimated to be 0.38, 0.57, 0.21, and 0.66 mol/mol, respectively. These yields are broadly similar to estimates derived by Ehhalt and Rohrer (2009) (0.37, 0.54, 0.19, 0.71, respectively) but are lower than estimates derived from box-model (0.38, 0.83, 0.38, and 0.85, respectively for $NO_x$=160 pptv (Grant et al., 2010)), which may reflect the impact of wet and dry deposition. In particular, Fig. S4 shows that the simulated yield of $H_2$ from $CH_4$ oxidation is lowest in the tropics, where most $CH_4$ is oxidized, as a greater fraction of $CH_2O$ is oxidized by OH in this region than at high latitudes.*

*Overall, we find that $CH_4$ oxidation is the largest photochemical source of $H_2$ ($\simeq 27$ Tg/yr). The oxidation of biogenic VOCs (BVOCs) accounts for the majority of the remaining photochemical source of $H_2$ ($\simeq 14$ Tg/yr) primarily from isoprene (8 Tg/yr), methanol (3 Tg/yr), and terpene (1 Tg/yr). The oxidation of VOCs from anthropogenic and biomass burning origin produces $\simeq 3$ Tg/yr of $H_2$. Our estimates are in good agreement with previous estimates (Ehhalt and Rohrer, 2009): $CH_4$ ($23 \pm 8$ Tg/yr), isoprene ($9 \pm 6$ Tg/yr), biomass burning and anthropogenic VOCs (3 Tg/yr). This similarity can attributed to the similar yield of $H_2$ from $CH_2O$ (0.4 mol/mol compared to 0.37 (Ehhalt and Rohrer, 2009)). More work is needed to better characterize the temperature and pressure sensitivity of $CH_2O$ photolysis quantum yields (Röth and Ehhalt, 2015).*

3. **Is a 9-year trend sufficient to carry out this analysis? Please note that the lifetime of H2, including soil sinks, is reasonably location dependent.**

The NOAA network provides much better spatial coverage than other networks and the fact that a statistically significant increase in H2 concentration is found at all background sites suggests that the increase in H2 is not associated with local processes. The 2010-2019 period is also especially interesting as Derwent et al. (2023) recently noted a statistically significant increase in $H_2$ at Mace-Head since 2010, which cannot be readily explained by known sources.

*The NOAA monitoring network provides additional spatial coverage that complements other existing networks (AGAGE (Prinn et al., 2018), CSIRO (Francey et al., 2003)) and offers a unique opportunity to evaluate the skill of the model in capturing global changes in $H_2$ atmospheric concentration since 2010. This period is especially important as recent $H_2$ observations at Mace Head (Derwent et al., 2021, 2023) show both an increase in $H_2$ concentration and its soil removal rate.*

4. **It is odd that the people who measured H2 and whose data is being recalibrated are not co-authors of this paper**
We apologize for the confusion. Gabrielle Pétron and Andrew Crotwell (second and third authors) are leading the recalibration of H2 observations at NOAA GML (Pétron et al., 2023) A paper describing the recalibration process in details is under review for publication in AMT (Pétron et al., submitted).

5. **Given that biogenic hydrocarbons such as isoprene are very short-lived, is the use of monthly climatologies appropriate? First, the yield of H2 from isoprene oxidation is not well known. Even the yield of H2 from formaldehyde photolysis has significant uncertainty. Second, it would depend very much on location and conditions.**

As indicated in the method section, isoprene (and terpene) emissions are calculated using the MEGAN inventory. We do not use a monthly climatology for these compounds and we have clarified that this calculation is performed interactively in the model.

*Biogenic emissions of VOCs are prescribed as a monthly climatology (Granier et al., 2005), except for isoprene and terpenes, of which emissions are calculated interactively using the Model of Emissions of Gases and Aerosols from Nature (Guenther et al., 2012).*

We agree with the reviewer that the yield of $H_2$ from isoprene will vary depending on the oxidative environment and the representation of the isoprene chemistry in the model. However, given the long lifetime of $H_2$, we believe that it makes sense to focus on the global yield of $H_2$, as it can be readily compared to other estimates. To facilitate such comparison, we have added estimates of both the overall source of $H_2$ from different VOCs and estimates of the molar yield of $H_2$ from their oxidation (see response to comment 2). In addition to changes made in response to comment 2, we have also added the following text to highlight the importance of the accurate representation of $CH_2O$ photolysis quantum yields.

*The oxidation of biogenic VOCs (BVOCs) accounts for the majority of the remaining photochemical source of $H_2$ ($\simeq 14$ Tg/yr) primarily from isoprene (8 Tg/yr), methanol (3 Tg/yr), and terpene (1 Tg/yr). The oxidation of VOCs from anthropogenic and biomass burning origin produces $\simeq 3$ Tg/yr of $H_2$. Our estimates are in good agreement with previous estimates (Ehhalt and Rohrer, 2009): $CH_4$ ($23 \pm 8$ Tg/yr), isoprene ($9 \pm 6$ Tg/yr), biomass burning and anthropogenic VOCs (3 Tg/yr). This similarity can attributed to the similar yield of $H_2$ from $CH_2O$ (0.4 mol/mol compared to 0.37 (Ehhalt and Rohrer, 2009)). More work is needed to better characterize the temperature and pressure sensitivity of $CH_2O$ photolysis quantum yields (Röth and Ehhalt, 2015).*

We have also clarified that the GFDL AM4.1 model uses FAST-JX to estimate the photolysis and quantum yields for $CH_2O$ photolysis.

*The production of $H_2$ associated with $CH_2O$ photolysis is calculated interactively using FAST-JX version 7.1, as described by Li et al. (2016).*

6. **Let us not forget that the soil uptake is a parameterization!! How good is it? How well is it checked out? (This whole issue is rather circular since the parameterization is not really a bottom-up approach that is tested.**

The two parameterizations that are used in the manuscript have been evaluated against observations previously in Paulot et al. (2021) and Bertagni et al. (2021). We have added a figure in the supporting materials summarizing the evaluation (see Fig. S6 below). While we agree with the reviewer that the representation of vd(H2) in all models carries significant uncertainty, the parameterizations used in this study provide a more mechanistic representation of H2 uptake than earlier approaches (Sanderson et al., 2003; Yashiro et al., 2011; Price et al., 2007). This is important as it allows us to perform more realistic exploration of the sensitivity to key uncertainties (e.g., HA-HOB moisture activation threshold). We have added the following text to the supporting materials:

*Fig. S6 shows the measured and simulated $v_d(H_2)$ at 7 different sites. As noted previously by Paulot et al. (2021), all parameterizations tend to overestimate the variability in $v_d(H_2)$ across sites. In particular, $v_d(H_2)$ is underestimated at Tsukuba and Mace Head (Fig. S6c and e). In contrast, seasonality and magnitude are well captured by all parameterizations at temperate sites (a, b, d, g). The large spread in simulated $v_d(H_2)$ at the San Jacinto Mountain Reserve (desert) reflects different degrees of inhibition of HA-HOBs under low soil moisture (Fig. S6f).*

7. **I do not understand how good are the trends shown in figure 4. What are the uncertainties?**

Fig. 4 shows the linear trends calculated using ordinary least square regression. In the preprint, the uncertainty in the estimated trend was denoted by an error bar. In the revised manuscript, we have further highlighted sites where the observed and simulated trends are significant (p<0.01) (see Fig. 4)

8. **The authors need to pay attention to significant figures.**
Thank you. We have addressed this issue throughout the manuscript (see for instance the revised budget figure (Fig. 1) above)

[Figure]

Figure 4: Trend in H$_2$ concentrations in observations (a) and in the BASE simulation (b) over the 2010–2019 period. Panel (c) shows the observed (black) and simulated (red) trend in H$_2$ at background sites (squares) as well as the trend in tagged H$_2$ tracers associated with anthropogenic sources (green), biomass burning (purple), ocean+soil sources (black), and photochemical production (blue). Filled symbols denote trends that are significantly different from 0 (p<0.01). The error bars show one standard deviation for the estimated observed and simulated trends.

Figure S6: Comparison between simulated and observed H$_2$ deposition velocity at (a) Harvard Forest (temperate forest Meredith et al. (2016)), (b) Gif-sur-Yvette (pasture Belviso et al. (2013), (c) Tsukuba (agricultural land Yonemura et al. (2000). (d) Helsinki (forest, Lallo et al. (2008)), (e) Mace Head (peat, Simmonds et al. (2011)) (f) San Jacinto Mountain Reserve (desert, Smith-Downey et al. (2008)) (g) Heidelberg (semi-urban, Hammer and Levin (2009))

---

## Author Response (AR2)

We thank the editor for accepting our manuscript. As requested, we have updated the color schemes used in your maps and charts as well as the "Competing interest statement".

We have also made the following two minor changes:

1. line 107: we corrected a small typo for the simulated molar yield of H2 from isoprene and terpene (0.56 and 0.70, respectively (and not 0.57 and 0.66)). We apologize for the mistake

2. line 126: we clarified that the increase in the photochemical source of H2 does not include non-methane VOCs from biomass burning and anthropogenic origins.